# A Low-Cost Modular Imaging System for Rapid, Multiplexed Immunofluorescence Detection in Clinical Tissues

**DOI:** 10.3390/ijms24087008

**Published:** 2023-04-10

**Authors:** Joshua Gu, Hannah Jian, Christine Wei, Jessica Shiu, Anand Ganesan, Weian Zhao, Per Niklas Hedde

**Affiliations:** 1Department of Biological Chemistry, University of California, Irvine, CA 92697, USA; 2Sue and Bill Gross Stem Cell Research Center, University of California, Irvine, CA 92697, USA; 3Department of Molecular Biology and Biochemistry, University of California, Irvine, CA 92697, USA; 4Department of Pharmaceutical Sciences, University of California, Irvine, CA 92697, USA; 5Department of Dermatology, University of California, Irvine, CA 92697, USA; 6Chao Family Comprehensive Cancer Center, University of California, Irvine, CA 92697, USA; 7Edwards Life Sciences Center for Advanced Cardiovascular Technology, University of California, Irvine, CA 92697, USA; 8Department of Biomedical Engineering, University of California, Irvine, CA 92697, USA; 9Institute for Immunology, University of California, Irvine, CA 92697, USA; 10Beckman Laser Institute and Medical Clinic, University of California, Irvine, CA 92697, USA

**Keywords:** immunofluorescence, cutaneous T-cell lymphoma, multiplex fluorescence imaging, in situ protein detection, tumor marker profiling

## Abstract

To image 4-plex immunofluorescence-stained tissue samples at a low cost with cellular level resolution and sensitivity and dynamic range required to detect lowly and highly abundant targets, here we describe a robust, inexpensive (<$9000), 3D printable portable imaging device (Tissue Imager). The Tissue Imager can immediately be deployed on benchtops for in situ protein detection in tissue samples. Applications for this device are broad, ranging from answering basic biological questions to clinical pathology, where immunofluorescence can detect a larger number of markers than the standard H&E or chromogenic immunohistochemistry (CIH) staining, while the low cost also allows usage in classrooms. After characterizing our platform’s specificity and sensitivity, we demonstrate imaging of a 4-plex immunology panel in human cutaneous T-cell lymphoma (CTCL) formalin-fixed paraffin-embedded (FFPE) tissue samples. From those images, positive cells were detected using CellProfiler, a popular open-source software package, for tumor marker profiling. We achieved a performance on par with commercial epifluorescence microscopes that are >10 times more expensive than our Tissue Imager. This device enables rapid immunofluorescence detection in tissue sections at a low cost for scientists and clinicians and can provide students with a hands-on experience to understand engineering and instrumentation. We note that for using the Tissue Imager as a medical device in clinical settings, a comprehensive review and approval processes would be required.

## 1. Introduction

Current cancer diagnosis methods are comprised of clinical examination, radiological imaging, and histopathological analysis of tissue biopsies and surgical resections, which provide insight into a patient’s type and stage of cancer [1,2]. Physicians have depended upon histopathology, which is the “gold standard” for visualization and pathological interpretation of tissue biopsies. Pathological analyses of tumor biopsies have broad utility in cancer diagnosis, prognosis, and treatment stratification. Hematoxylin and eosin (H&E)-stained histologic sections are considered the gold standard by pathologists and can be used for a variety of applications, such as identifying malignant tumors, segmentation of glands in the prostate, grading of breast cancer pathology, and classification of early pancreatic cancer [3]. The immunohistochemistry (IHC) method, chromogenic immunohistochemistry (CIH), is used to complement H&E staining, which stains the tissue morphology, to detect the presence of specific protein markers for accurate tumor classification and diagnosis. While H&E and CIH stains provide enough information for some applications, there are many cases, such as tumor differentiation and tumor immune microenvironment (TIME) profiling, where more data are needed. In addition, conventional CIH is limited to a few markers per tissue section and chromogenic systems used for the staining saturate easily, restricting quantitative analysis [4,5]. In these cases, labeling the cells with antibodies for immunofluorescence imaging can allow for multiplexing, increase the sensitivity and dynamic range, and provide additional information for further characterization [3,6,7]. Even though immunofluorescence provides clinical value, it currently requires expensive imaging hardware, and the acquisition of a large field of views to generate sufficient data can be very time-intensive.

The ability to multiplex immunofluorescence markers enables studies that investigate cellular co-expression [8,9], cellular spatial relationships [10], and tissue heterogeneity [11], to name a few. In the field of immunotherapy, understanding the cellular composition and spatial distribution within the sample, which is referred to as spatial biology, has become important [5,12,13]. By profiling immune checkpoint inhibitors, which reduce T-cell inhibition and allow them to fight cancer cells, cancer treatments have benefited [14,15,16,17,18,19].

Cutaneous T-cell lymphoma (CTCL) is a type of cancer that starts in white blood cells called T cells (T lymphocytes), which typically help fight pathogens in the immune system [20]. In CTCL, T cells develop abnormalities, causing them to attack the skin and cause rash-like skin erythema, patches of raised or scaly skin, and sometimes skin tumors [21]. Unfortunately, the exact cause of CTCL is still unknown. As CTCL tissue samples contain high levels of T cells, they are a good positive control for T-cell markers such as CD3, CD8, and CD14 [22]. Hence, we selected CTCL tissue samples to be our model system to demonstrate detection of these T-cell markers, which vary from low to high abundance to demonstrate the sensitivity of our imaging platform.

To take full advantage of the clinical value of immunofluorescence, a robust, inexpensive, high-throughput imaging platform that can be deployed immediately to any laboratory or clinic, including those in low-resource settings to image clinical tissue samples with immunofluorescence, is highly desired. To address this need, we have developed a robust, inexpensive (<$9000), and portable imaging platform for tissue samples, the Tissue Imager, that can be placed on the benchtop of any basic laboratory. Our Tissue Imager uses a 3D printable design and widely available components to excite fluorescence of fluorophore-conjugated secondary antibodies that are detected with an inexpensive 20-megapixel CMOS camera module coupled with a long working distance, 10× objective, with sufficient spatial resolution to provide cellular resolution and sensitivity to detect a wide range of protein abundance levels. We demonstrate that, with clinical patient samples, this imaging platform can obtain image resolutions or par with a commercial epifluorescence microscope that is >10 times more expensive while, at the same time, providing a larger field of view to increase imaging throughput. Our low-cost, high-throughput, and portable platform can immediately benefit the scientific community, and eventually, the healthcare community as well.

## 2. Results

### 2.1. System Design

While existing low-cost microscopy platforms designed for biological fluorescence utilize the camera of cellphones with various illumination schemes such as on-axis epi-illumination [23,24], off-axis inclined illuminations [25], butt-coupling [26], and total internal reflection [27], there have been limitations regarding spatial resolution, field of view, and the maximum number of spectral channels. To obtain sub-cellular spatial resolution (~1 μm) and multiplexed fluorescence images for clinical tissue biopsy samples mounted on glass microscope slides (25 mm × 75 mm × 1 mm), a device would need to feature an objective lens of reasonably high numerical aperture (NA~0.3) as well as multiple spectral windows for illumination and detection of several different fluorophores. For the imager to be inexpensive and able to image samples in a high-throughput manner, it should be portable, low-cost, and easy to use by technicians with minimal training. Our Tissue Imager meets all these requirements to image tissue samples with multiplexed immunofluorescence staining, as illustrated in Figure 1A–C. The tissue samples are first stained with antibodies, then imaged with the Tissue Imager, followed by analysis. The overall design of the Tissue Imager can be seen in Figure 1D, with an overall dimension of 25 cm × 25 cm × 42 cm. A photograph of the assembled device is shown in Figure 1E. Clinical tissue sample sections from pathology centers are typically also placed on glass microscope slides of 25 mm × 75 mm, hence our Tissue Imager was designed to accommodate this format on the sample stage. The images obtained with a 20-megapixel CMOS camera (3648 × 5472 px) correspond to a 1.8 × 2.6 mm^2^ field of view (FOV) with a pixel size at the sample of 0.48 μm (Appendix A), sufficiently large enough to image a typical human biopsy section such as skin tissue. To validate and benchmark our Tissue Imager, we acquired reference images with a Nikon Ti-1000E widefield microscope using a 10× objective with a similar sample pixel size of 0.65 μm.

Tissue sample images were acquired from the top with a 10×-long working distance objective (Mitutoyo Plan Apochromat, NA 0.28), followed by a six-position motorized filter wheel (five bandpass filters currently used with center wavelengths 460 nm, 530 nm, 577 nm, 645 nm, and 690 nm) to spectrally select the fluorescence emission from each fluorophore type on the sample. The fluorescence was then imaged with a tube lens (f = 100 mm) onto the chip of a 20-megapixel CMOS camera and read out via a USB 3 interface compatible with most computers and operating systems. Fluorescence excitation was achieved using a ring-like structure above the sample that held five LEDs, each coupled to a condenser lens (f = 20 mm) and cleanup filter (center wavelengths 365 nm, 460 nm, 520 nm, 585 nm, and 630 nm). The sample was placed below onto an xyz sample stage allowing for field-of-view position and focus adjustments. The entire setup was enclosed in a box made from black ¼”-thick laser-cut acrylic boards. This light-tight enclosure prevented external light from contaminating the resulting images. All components including the optics, camera, LEDs, and structural supports were integrated into a CAD model that could be manufactured on a larger scale at a low cost. To adjust and evaluate the illumination homogeneity, images of a reference microscope glass slide were taken for each channel. The remaining variations could be easily corrected through software, as seen in the H&E image obtained as an RGYB image (Appendix A). In addition to fluorescence imaging, our device also allowed for the acquisition of brightfield images as required for IHC- or H&E-stained samples. For this purpose, separate images with red (630 nm), green (520 nm), yellow (577 nm), and blue (460 nm) illumination were taken and merged into a final RGYB color image.

### 2.2. Evaluation of Specificity and Sensitivity

To evaluate the performance of the Tissue Imager, we imaged fluorescence beads of various emission/detection ranges to validate all five spectral channels. After vortexing and diluting each 1 µm FluoSpheres™ Polystyrene Microspheres (blue/green, yellow/green, orange, red, and crimson) sample 1:2000 in PBS, 10 µL of sample solution was pipetted into Countess™ Cell Counting Chamber Slides (Invitrogen, Waltham, MA, USA, C10228). As shown in Figure 2, each microsphere population was detected in the expected spectral channel. To evaluate potential spectral crosstalk between channels, each microsphere population was imaged in all channels. We found the fluorescence to be specific to the respective channels, demonstrating the specificity of the Tissue Imager and its ability to resolve beads as small as 1 µm in diameter (Appendix A).

To determine the sensitivity of the Tissue Imager, the Dragon Green (DG) intensity standard beads (Bangs Laboratories, Fishers, IN, USA, DG06M) of five different intensities (DG1-DG5) were imaged. This standard bead kit is typically used for fluorescence microscopy and flow cytometry calibrations. The standard beads were vortexed and diluted 1:10 in PBS-T (0.025% Tween20), then 10 µL of sample solution was pipetted into Countess™ Cell Counting Chamber Slides (Invitrogen, Waltham, MA, USA, C10228) for imaging. As shown in Figure 3A, the beads were excited with the 460 nm LED and detected in the 530 nm channel with fluorescence intensities increasing from DG1 to DG5 as expected. In Figure 3B, the fluorescence intensity for each bead intensity was quantified using ImageJ [28,29] and plotted to characterize the sensitivity and wide dynamic range (0.24–100% intensity) of the Tissue Imager.

### 2.3. Evaluation of an Immune Panel on CTCL Tissue Samples

The next step was to profile immune markers in clinical tissue samples to demonstrate rapid imaging for a 4-plex protein detection panel. Using our CTCL model, we profiled CD3e, CD8, and CD14 using antibodies. CD3e and CD8 are T-cell markers, while CD14 has been used as a marker for monocytes and macrophages [22,30,31]. The nucleus was stained with DAPI. The images obtained from the Tissue Imager were compared to H&E and CD3e and CD8 IHC stains from serial sections of the same FFPE block. The CD3e and CD8 from the same section imaged on the Tissue Imager were also imaged on a Nikon Ti-1000E microscope with a 10× objective lens as a benchmark for immunofluorescence imaging (Figure 4). The images shown in Figure 4A,B were representative images of a total of seven serial sections that were stained, imaged, and analyzed. CD14 was not imaged on the Nikon microscope due to the absence of a suitable spectral channel. We note that we focused on the T-cell markers CD3e and CD8, which are more commonly used in studies of cutaneous T-cell lymphoma (CTCL). The absence of CD14 staining in the Nikon images does not invalidate the results of the study, as we were still able to demonstrate the Tissue Imager’s ability to detect multiple markers simultaneously and compare its performance to a conventional microscope for CD3e and CD8 staining. As confirmed by the IHC staining (kindly provided by the UCI Dermatology Center), CD3e is highly abundant, while CD8 is less abundant (Figure 4C). This allowed us to demonstrate the Tissue Imager’s ability to detect protein markers of various abundance levels. The DAPI- (405 nm), CD3e- (488 nm), CD14- (594 nm), and CD8- (647 nm) stained CTCL tissue section was imaged within six seconds on the Tissue Imager. As seen in Figure 4D, the intensity of twelve randomly selected cells positive in each channel were measured along with the local background to compare the signal-to-noise ratio (SNR) between images acquired on the Tissue Imager and Nikon. As negative control, tissues were stained with the secondary antibody only (Appendix A). A Bland–Altman plot of the SNR differences between the Tissue Imager and the Nikon microscope is shown in Appendix A.

After acquisition, the images were processed using ImageJ and analyzed using a CellProfiler [32,33] image analysis pipeline (Figure 5A). The CellProfiler pipeline was validated by manually counting six 700 × 700 px regions of interest that were randomly selected throughout the tissue section. The manual counting was used to obtain the percentage of cells positive for each marker and compared to the counts detected in the CellProfiler pipeline. As shown in Appendix A, there were no significant differences between the manual counts and CellProfiler counts for all three markers (CD3e, CD8, and CD14). By detecting the DAPI-stained nuclei, 8238 cells were found in this image (Figure 5B). The cells positive for each marker were then detected and quantified, with 51% of cells expressing CD3e, 16% of cells expressing CD8, and 18% of cells expressing CD14 (Figure 5C). The percentages of cells positive for each marker were then plotted for all images (*n* = 7), resulting in an average of 49%, 15%, and 12% cells positive for CD3e, CD8, and CD14, respectively (Figure 5D). The CellProfiler pipeline also detected cells that co-expressed both CD3e and CD8. On average, 9% of cells were CD3e/CD8-positive, 40% of cells were CD3e-positive/CD8-negative, and 5.7% of cells were CD3e-negative/CD8-positive (Figure 5E).

## 3. Discussion

Here we have demonstrated that our Tissue Imager can achieved an imaging performance on par with commercial epifluorescence microscopes for imaging of a 4-plex immunology panel in human CTCL FFPE tissues. The ability to detect co-expression of multiple protein markers in the same cell at the single-cell level is of high relevance in clinical pathology, particularly for profiling of both the presence of T cells and the abundance of immune checkpoint proteins for patient stratification. The compatibility of Tissue Imager data with an automated marker counting pipeline underscores the capabilities of this device.

The main limitation of this study was the relatively small sample size. The study only used tissue samples from a CTCL model, which may not represent the full range of tissues and diseases. A larger sample size and diverse tissue samples may be needed to further validate the findings. Further, as an epifluorescence microscope, the Tissue Imager has a limitation of requiring thin slicing of tissue samples, which means that samples with a thickness greater than 10 µm cannot be imaged effectively. This limitation arises because epifluorescence microscopy lacks optical sectioning capabilities, which leads to high background noise when thicker tissues are imaged. Also, in its current form, the Tissue Imager does not have an objective turret, which means that magnification cannot be changed during imaging. Another limitation of the current setup is the lack of a white light source and RGB filters needed to reproduce the spectral response of a color camera, which limits its use for imaging stains such as H&E and IHC. The current method is to sequentially image with excitation and detection settings E630 nm/D645 nm, E585 nm/D577 nm, E520 nm/D530 nm, and E460 nm/D460 nm to sequentially acquire the red, yellow, green, and blue (RYGB) portions of a brightfield image (Appendix A). This approach can emulate brightfield imaging, but some color variations can be seen compared to a true brightfield image (Appendix A) due to the narrow filter bandwidths. The microscope also has limited sensitivity due to the usage of a relatively low numerical aperture objective, which would render it hard to image targets with low copy numbers such as RNA. Additionally, the Tissue Imager is currently not automated, and a motorized stage would be required for high-throughput imaging applications. Finally, if the Tissue Imager is to be used as a medical device in clinical settings, it would require a comprehensive review and approval process to ensure that it meets the necessary regulatory requirements.

In the future, some of these limitations could be addressed by incorporating additional features such as additional spectral channels and potentially hyperspectral detection, a motorized/automated sample stage and objective turret, and possibly even fluorescence lifetime detection with time-of-flight-resolving consumer cameras. With additional spectral channels, hyperspectral detection, and/or fluorescence lifetime imaging microscopy (FLIM), users would be able to multiplex higher and remove autofluorescence from images with FLIM analysis. With oligo-conjugated antibodies, users can multiplex above 3–4 plex in a single round of staining and imaging and possibly use combinatorial labeling and FLIM to multiplex and perform decoding to improve detection accuracy. Additionally, the light path could be modified to include a broadband light source and (phase) masks to enable dark-field and phase-contrast imaging. Fluorescence quenchers such as TrueBlack could be used to quench tissue autofluorescence, thus increasing sensitivity, and novel computational tools could be leveraged to maximize the information obtained from the images [34]. Specifically, the use of deep learning-based image analysis approaches could improve the accuracy of cell segmentation in microscopy images [35].

## 4. Materials and Methods

### 4.1. Tissue Imager Design

The sample slides were illuminated with five different LEDs (365 nm, 460 nm, 520 nm, 585 nm, 630 nm, 120° angle of emission) using a custom-designed 5-channel LED ring mounted above the sample stage. The LEDs were driven with constant current LED drivers, resulting in output powers of 150 lm (460 nm), 115 lm (630 nm), 500 lm (520 nm), and 500 lm (585 nm). After collimation with aspherical lenses of 20 mm focal length (Thorlabs, Newton, NJ, USA), the LED emissions were spectrally cleaned with bandpass filters (365/10 nm, 460/30 nm, 520/20 nm, 585/20 nm, 630/20 nm) (Chroma, Bellows Falls, VT, USA). LEDs were driven by individual DC–DC driver circuits to adjust the current (max 1000 mA each). Fluorescence was collected with a long working distance (WD 34 mm) 10× Mitutoyo Plan Apochromat Objective (Thorlabs, Newton, NJ, USA) coupled to a 1” diameter achromatic tube lens of 100 mm focal length (Thorlabs, Newton, NJ, USA). The custom six-position filter wheel was actuated with a servo motor controlled with an Arduino Nano microcontroller, that was also used to control the LED drivers. Before imaging with a 20 MP monochrome CMOS camera (FLIR Blackfly, FLIR Systems, Goleta, CA, USA), bandpass filters were used to block scattered excitation light (450/50 nm, 530/30 nm, 577/25 nm, 645/30 nm, 690/50 nm) (Chroma, Bellows Falls, VT, USA). All electronics were powered by a 5 V, 3.5 A power supply. After 3D printing of the model (Ultimaker S5, Ultimaker B.V., Utrecht, The Netherlands), all relevant optical components were inserted and attached.

### 4.2. Optical Resolution Characterization

We have taken cross sections of multiple 1 µm fluorescent beads to characterize the optical resolution of our imager.

### 4.3. Pixel Size Calibration Measurements

A 10 mm ruler (R1L3S1P, Thorlabs, Newton, NJ, USA) was imaged with RGB settings for a brightfield image. Since we used a monochrome camera with filters, we emulated RGB image acquisition by sequentially illuminating with blue (460 nm), green (530 nm), and red (630 nm) light and acquisition in the corresponding channels. The image was then quantified using ImageJ by measuring the distance in pixels between 1 division (50 µm) or 2 divisions (100 µm). The µm/pixel value was then calculated, and the average value was obtained (Appendix A).

### 4.4. Fluorescent Beads

Then, 1 µm FluoSpheres™ Polystyrene Microspheres of various colors (Invitrogen, Waltham, MA, USA, F13080, F13081, F13082, F13083, F8816) were vortexed and diluted at 1:2000 with PBS before being pipetted into a Countess™ Cell Counting Chamber Slide (Invitrogen, Waltham, MA, USA, C10228). The Dragon Green Intensity Standard Kit with 5 Intensities (Bangs Laboratories, Fishers, IN, USA, DG06M) (DM1-5) around 8 µm diameter were vortexed, diluted 1:10 in PBS-T (0.025% Tween20), and pipetted into a Countess™ Cell Counting Chamber Slide (Invitrogen, Waltham, MA, USA, C10228).

### 4.5. Preparation of FFPE Tissues

The University of California Irvine IRB approved this study for IRB exemption under protocol number HS# 2019-5054. All methods were carried out in accordance with relevant guidelines and regulations. All human cutaneous T-cell lymphoma (CTCL) cases were de-identified samples to the research team at all points and therefore considered exempt for participation consent by the IRB. Fully characterized human patient skin CTCL FFPE tissues were achieved samples obtained from the UCI Dermatopathology Center, then sectioned to 5 µm-thick slices using a rotary microtome, collected in a water bath at 35 °C, and mounted to positively charged Fisher super frost coated slides (Fisher Scientific, Waltham, MA, USA, 12-550-15). The tissue sections were then baked at 60 °C for 1 h. For antigen unmasking, slides were deparaffinized, rehydrated, then followed by target retrieval (with citrate buffer).

### 4.6. Antibody Staining

The samples were blocked with 10% BSA in PBS for 2 h at room temperature. Antibody solutions containing Rabbit anti-Human CD3e (Abcam, Cambridge, UK, ab52959), Mouse anti-Human CD8 (Abcam, Cambridge, UK, ab75129), and Goat anti-Human CD14 (LifeSpan, Providence, RI, USA, LS-B3012-50) antibodies and 1% BSA in PBS were subsequently added to the samples and incubated overnight at 4 °C. Following a PBS wash, antibody solutions containing fluorescently labeled Donkey anti-Rabbit Alexa 488 (Thermo Fisher, Waltham, MA, USA, A-21206), Donkey anti-Mouse Alexa 647 (ThermoFisher, Waltham, MA, USA, A-31571), and Donkey anti-Goat Alexa 594 (Thermo Fisher, Waltham, MA, USA, A32758) antibodies in 5% secondary raised serum and 1% BSA in PBS were added at room temperature for 1 h. Three 5 min washes at room temperature with RNAse-free PBS were then performed, with the second wash containing 1:1000 Hoechst stain.

### 4.7. Image Acquisition and Data Transfer

Note that 1 µm fluorescence beads were imaged with a camera exposure time of 1000 ms at 365/460 nm excitation/emission, and a camera exposure time of 100 ms was used for all remaining channels. The Dragon Green Intensity Standard Kit (Bangs Laboratories, Fishers, IN, USA, DG06M) was imaged with an exposure time of 1000 ms in the 460/530 nm channel (excitation/emission). Tissue sections were imaged with an exposure time of 300 ms for DAPI staining (excitation 365 nm, detection 460 nm), 1500 ms for the 488 nm (excitation 460 nm, detection 530 nm) and 647 nm (excitation 630 nm, detection 690 nm) channels, and 2000 ms for the 594 nm (excitation 585 nm, detection 645 nm) channel. For all images taken, the camera gain was set to 26 dB. Images were saved in 16-bit TIFF format for further processing. For analysis, the tissue images were cropped 2300 × 2300 px.

Validation images were acquired with an inverted Nikon Ti-1000E epifluorescence microscope using a 10× plan apochromat oil objective with a numerical aperture of NA 0.45. Samples were excited with a Spectra-X (Lumencor, Beaverton, OR, USA) LED light source at 395 nm (200 ms; 5% laser power), 470 nm (200 ms; 25% laser power), and 640 nm (200 ms; 50% laser power). Images were acquired with an Andor Zyla 4.2 sCMOS camera. The H&E image in Appendix A was taken on a Nikon Eclipse E400 with the Nikon Plan Fluor 10×/0.30 DIC objective lens and a QImaging MicroPublisher 6 camera.

### 4.8. ImageJ Image Processing

The open-source software ImageJ was used to pseudocolor the images from each channel acquired on the Tissue Imager (E365 nm/D460 nm: blue; E460 nm/D530 nm: green; E520 nm/D577 nm: yellow; E585 nm/D645 nm: magenta; E630 nm/D690 nm: red). The channels were then merged to generate a merged image. Scale bars were generated using the measurement of 0.48 μm/px. The lookup tables (LUTs) were also adjusted in ImageJ with the same setting across all images that were compared.

The SNR (signal-to-noise ratio) was calculated by dividing the intensity value of the protein by that of the background. Protein and background intensity values were averaged for each FOV (field of view). The percentage of positive cells was calculated by dividing the number of cells positive for the protein of interest by the total number of cells detected in the FOV.

### 4.9. CellProfiler

Fluorescence signal intensity was quantified using the open-source software CellProfiler. Raw *.nd2 images from Nikon and composite Tissue Imager images created with another open-source software, ImageJ 1.53c, were fed into a CellProfiler pipeline. In the pipeline, the nuclei were identified using the “IdentifyPrimaryObjects” module then expanded to represent the cell bodies. Protein fluorescence was also identified with the “IdentifyPrimaryObjects” module. Raw channel images were rescaled with the “RescaleIntensity” module for accurate protein and background intensity measurements that were obtained using the “MeasureObjectIntensity” and “MeasureImageIntensity” modules, respectively. Positive cell determination was done using the “RelateObjects” module.

### 4.10. Manual Counting

To validate the CellProfiler pipeline, positive cells were manually counted with the Cell Counter plugin on the open-source software ImageJ. Tissue Imager images were cropped to 700 × 700 px with a total of six fields of view in various regions of the sample. The DAPI channel and the fluorescent channel labeling the protein of interest were merged using ImageJ. Any cell with the fluorescent signal indicative of the presence of the protein was manually marked as a positive cell with a single dot in the image and counted by the Cell Counter. The percentage of cells positive obtained via manual counting was then compared to the percentage of cells positive from the CellProfiler pipeline.

### 4.11. Statistical Analysis

Student (two-sided) *t*-tests were performed for the comparison between manual counts and CellProfiler counts. For Figure 3, each fluorescent bead population had 13 beads selected randomly throughout the image for quantification on ImageJ. For Figure 4 and Figure 5, a total of 7 serial sections of CTCL tissue were stained, imaged, and analyzed. A total of 12 cells in each channel were randomly selected for fluorescence quantification through ImageJ for Figure 4D. In Figure 4D, the *p* value was >0.11 for the 488 nm channel and classified as not significant (n.s.). For Appendix A, the *p* values for the student *t*-test between counting methods were >0.75, >0.82, and >0.22 for CD3e, CD8, and CD14, respectively.

### 4.12. Workflow Overview

A schematic overview of the workflow described above is depicted in Appendix A.

## 5. Conclusions

In summary, the Tissue Imager described here represents a low-cost instrument (<$9000) which is a simple yet sensitive and highly versatile (five fluorescence channels + RYGB brightfield) design that could be reproduced easily, thus being a useful tool in settings such as academic laboratories. This device provides a low-cost platform for scientists to rapidly image clinical samples on lab benchtops or any location with little space available as well as an opportunity for students to gain the knowledge and experience in engineering, instrumentation, and software development. Basic analysis modules are also available on ImageJ, providing users with the opportunity to learn about these algorithms and create their own Tissue Imager workflows. This device could also be used for other applications such as tissue microarray imaging with minimal modifications to enable high-throughput batch sample analysis.

## Figures and Tables

**Figure 1 ijms-24-07008-f001:**
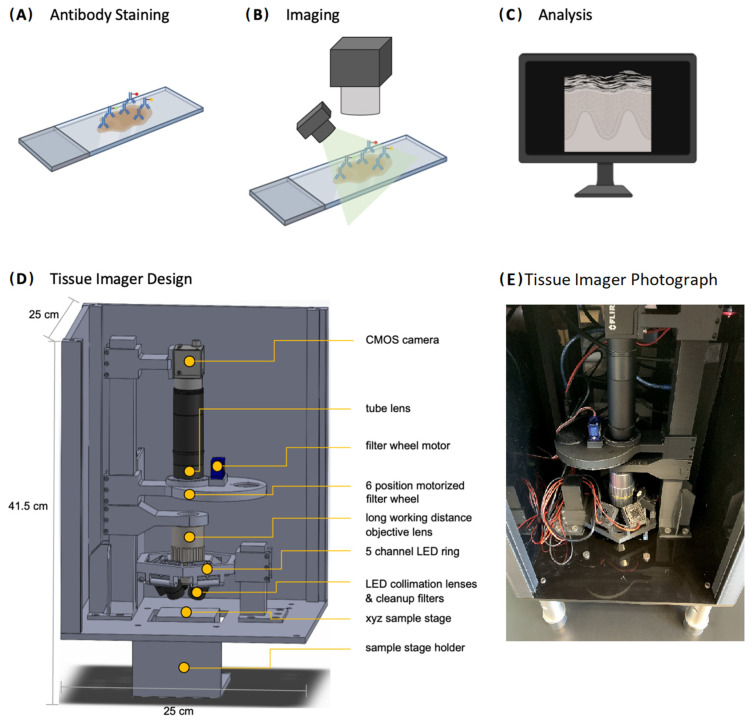
Workflow/design schematic. (**A**) Tissue samples are processed, and the protein markers are stained with antibodies. After blocking (e.g., typically with 10% BSA in this study), the primary antibody cocktail is placed onto the sample, followed by incubation with secondary labeled antibodies. (**B**) The sample is then imaged with the Tissue Imager. (**C**) The images are saved into a folder automatically and the images are then exported into ImageJ and/or CellProfiler for processing and quantification. (**D**) Tissue Imager CAD design with 20-megapixel CMOS camera, motorized filter wheel, 10× objective lens, five excitation LEDs, and sample stage holder. (**E**) Photograph of the Tissue Imager with enclosure panel removed.

**Figure 2 ijms-24-07008-f002:**
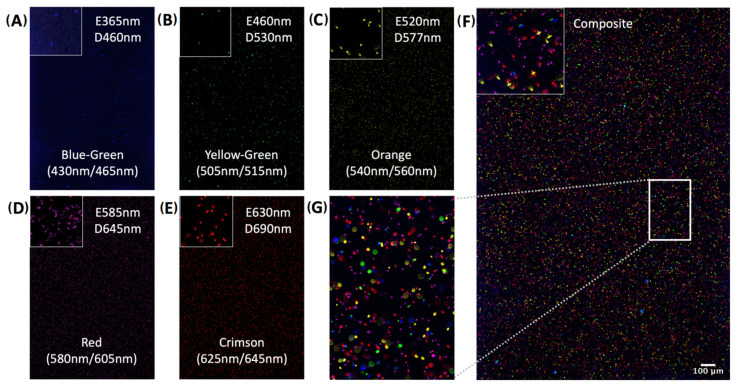
Validation of each spectral channel and the specificity of the Tissue Imager. Validation of each spectral channel and the specificity of the Tissue Imager. Using 1 µm fluorescent beads at a 1:2000 dilution, each spectral channel was calibrated. (**A**) Blue-green fluorescence beads with excitation/emission peaks at 430 nm and 465 nm illuminated at 365 nm and detected at 460 nm. (**B**) Yellow-green fluorescence beads with excitation/emission peaks at 505 nm and 515 nm, illuminated at 460 nm and detected at 530 nm. (**C**) Orange fluorescence beads with excitation/emission peaks at 540 nm and 560 nm, illuminated at 520 nm and detected at 577 nm. (**D**) Red fluorescence beads with excitation/emission peaks at 580 nm and 605 nm, illuminated at 585 nm and detected at 645 nm. (**E**) Crimson fluorescence beads with excitation/emission peaks at 625 nm and 645 nm, illuminated at 630 nm and detected at 690 nm. (**F**) Composite image of all channels. (**G**) Inset of composite image F. Scale bar, 100 µm.

**Figure 3 ijms-24-07008-f003:**
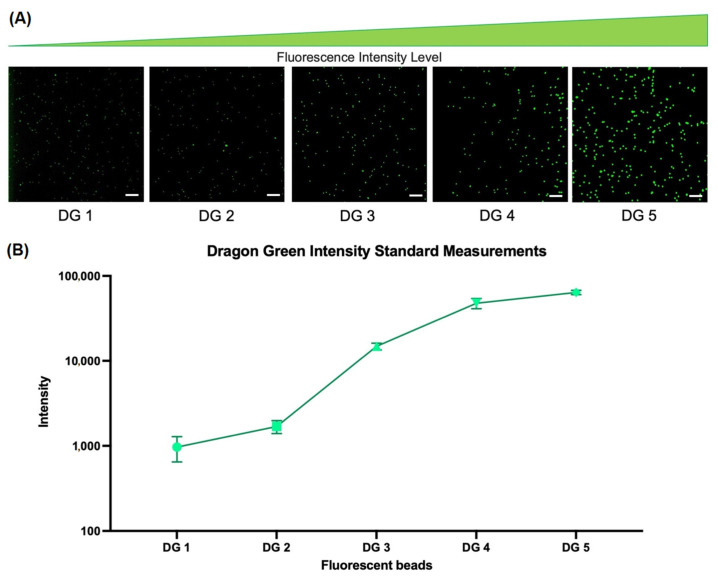
Quantifying sensitivity of the Tissue Imager. Quantifying sensitivity of the Tissue Imager. (**A**) Image of the Dragon Green (DG) intensity standard beads of ~8 µm diameter polystyrene-based microspheres dyed with increasing amounts of the DG fluorophore to quantify the sensitivity of the Tissue Imager. (**B**) Average intensity values of each bead population are plotted. The scatter plot shows the average values with standard deviations. Scale bars, 100 µm.

**Figure 4 ijms-24-07008-f004:**
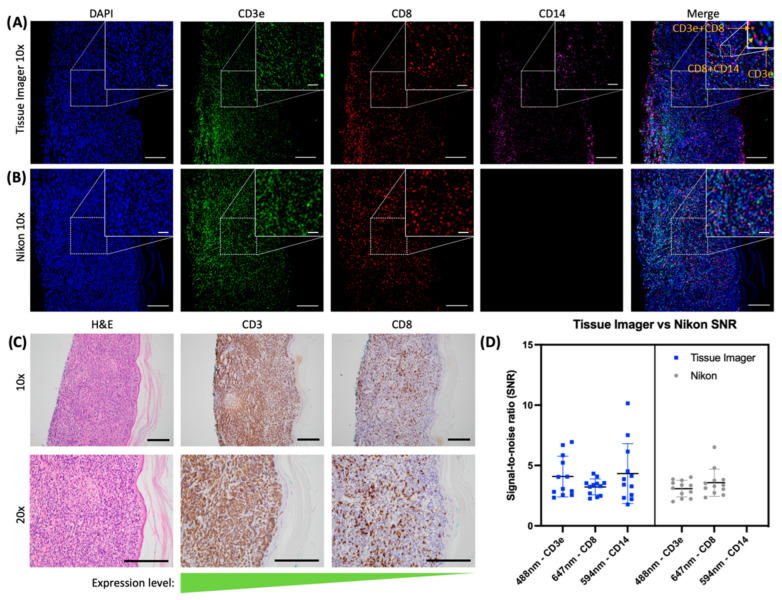
A 4-plex immune panel on CTCL with comparison to IHC and Nikon microscope. A 4-plex immune panel on CTCL with comparison to IHC and Nikon microscope. Immunofluorescence staining of DAPI, CD3e, CD8, and CD14 on Cutaneous T-cell Lymphoma (CTCL) FFPE tissue sections imaged with the Tissue Imager and Nikon. Images were compared to H&E staining, CD3 IHC, and CD8 IHC of serial sections. (**A**) Tissue Imager 10× images cropped for channels (DAPI, CD3e, CD8, CD14) and merged. Example protein puncta are indicated by arrows in the merged image. (**B**) Nikon Ti-1000E 10× images for channels (DAPI, CD3e, CD8) and merged. (**C**) 10× and 20× images of H&E, CD3 IHC, and CD8 IHC staining on serial sections of CTCL FFPE tissue sections. Brown staining is a positive signal. (**D**) Signal-to-noise ratio (SNR) comparison between Tissue Imager and Nikon images for each channel (*n* = 12). Scatter plots were plotted with mean and standard deviation. A student *t*-test was performed for the 488 nm channel between the Tissue Imager and Nikon Ti-1000E had a *p* value of >0.11 (n.s.). Scale bars, 200 µm for large images and 50 µm for insets.

**Figure 5 ijms-24-07008-f005:**
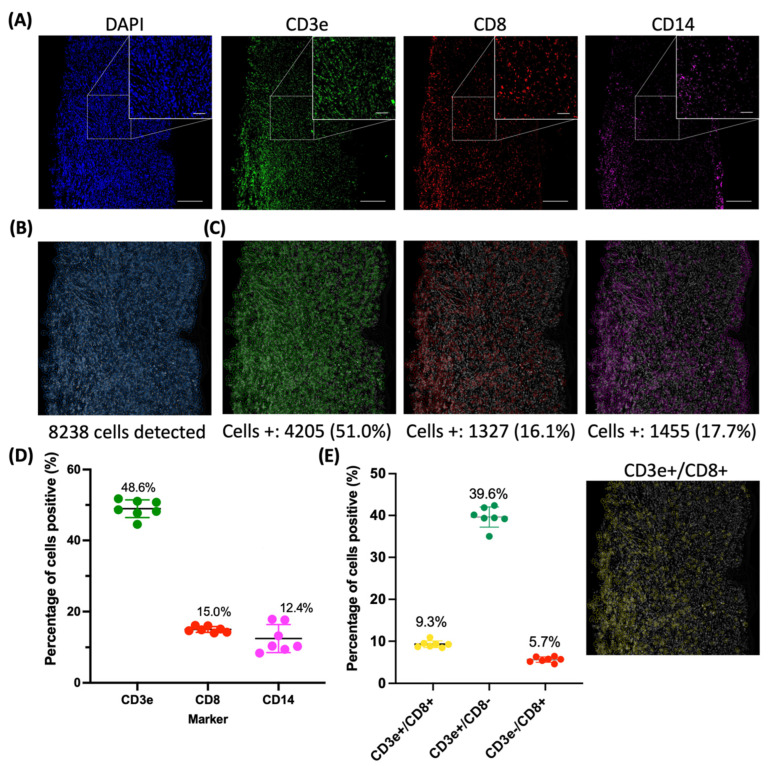
CellProfiler quantification of images and cell segmentation. CellProfiler quantification of images and cell segmentation. (**A**) The fluorescence images obtained from the Tissue Imager (DAPI, CD3e, CD8, CD14). (**B**) The cell segmentation outline from the CellProfiler pipeline. (**C**) Cells identified as positive for the staining for each marker were outlined and the cells-positive percentages were calculated (*n* = 7). (**D**) Percentage of cells positive for each marker with the average of 49%, 15%, and 12% for CD3e, CD8, and CD14, respectively. (**E**) Percentage of cells CD3e+/CD8+, CD3e+/CD8−, CD3e−/CD8+ were, on average, 9.3%, 40%, and 5.7%, respectively. Outline of cells CD3e+/CD8+. Scatter plots were plotted with the mean and standard deviation.

## Data Availability

The datasets generated during and/or analyzed during the current study are available from the corresponding author on reasonable request.

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
