# Peer review of "A Low-Cost Modular Imaging System for Rapid, Multiplexed Immunofluorescence Detection in Clinical Tissues"

_ijms, 2023, doi:10.3390/ijms24087008_

Round 1

Reviewer 1 Report

The manuscript presents an interesting and useful low cost modular imaging system for rapid multiplexed immunofluorescence detection in clinical tissues. 

However the paper design requires a major revision: 

1. the results section includes the information about the proposed imaging system. It should be described in a separate section as the system design and/or study design;

2. the material and methods should be included before the results. It could be useful to provide a schematic description (e.g. a diagram), step be step, how the concrete evaluation should be  performed;

3. Then, next part of the manuscript: results, discussion and conclusions. Unfortunately, conclusions are missing, moreover key words are missing too.

Author Response

Please refer to attached response letter.

Reviewer 2 Report

Dear Editor,

I am sharing my review of the ijms-2275499 entitled: A LOW-COST MODULAR IMAGING SYSTEM FOR RAPID, MULTIPLEXED IMMUNOFLUORESCENCE DETECTION IN CLINICAL TISSUES.

Current cancer diagnosis methods include clinical examination, radiological imaging, and histopathological analysis of tissue biopsies. Hematoxylin and eosin (H&E) stained histologic sections are considered the gold standard by pathologists, and the immunohistochemistry (IHC) method is used to detect specific protein markers for accurate tumor classification and diagnosis. In cases where more data is needed, labelling cells with antibodies for immunofluorescence imaging can allow multiplexing and provide additional information for further characterization. A robust, inexpensive, high-throughput imaging platform for immunofluorescence is highly desired. The authors have developed a portable platform to benefit scientific and healthcare communities. Cutaneous T cell lymphoma (CTCL) tissue samples containing high levels of T cells were used as a model system to demonstrate the detection of T cell markers. The paper needs a minor revision due to the following point:

·         Figure 1 add an image of the actual Tissue imager your lab next to the schematic image

·         Figure 3 and Figure 4 A metering bar description is missing

·         For Figures 4 A and B, please use images with the same high magnification for both imaging methods and indicate details of the merged sample DAPI CD3e, CD8 CD14.

·         Maybe you should also add a Bland-Altman plot to compare two methods (see also https://doi.org/10.1016/j.exphem.2020.09.191, https://doi.org/10.1016/j.saa.2022.121092, https://doi.org/10.1016/j.saa.2022.121940)

·         Maybe you should also use QuPath for the digital image analysis of the markers (https://qupath.github.io/).

·         Is it possible to provide a link to a detailed description of the Tissue Imager design with the cad files for 3D printing?

Thanks for this impressive manuscript.

Best

Reviewer 3 Report

In this work the authors describe a novel low-cost imaging system for the multiplexed fluorescent acquisition of samples of clinical interest. In particular, the tissue imager system is cheaper compared to conventional microscopy and can be easily used on benchtops for the detection of fluorescent proteins. In this study, tissue imager is used to analyse samples of human cutaneous T-cell lymphoma, showing a good specificity and reliability. The work is original and well presented as well as the used methodology, and the language is rather good. However, I have some major concerns listed below that should be carefully addressed by the authors to have their paper acceptable for publication.

Minor

·      Pag1 line33. Please, specify the meaning of “FFPE” at the first mention 

·      Figure 4 panel C. The authors should add the scale bar for images of H&E and IHC.

·      Figure 5. In panel B total detected cells are reported with comma, while in the panel C they don’t. Please uniform this point. 

·      Figure 5 panel D. It is useless to me to have the name of the markers both on the legend with colored dots and on the x-axe.  So, I suggest to remove the legends with colored dots or on the x-axe in order to make the graph lighter.

Major

·      Figure 4. Why did the authors not report the H&E and IHC of CD14 marker in the panel C as done for CD3 and CD8? It is clear that this was not reported with the conventional microscope due to the absence of suitable spectral channel but it could be shown for histological sections. Please discuss this point.  

·      The discussion section is rather poor. According to me, the authors should also debate about the limits of the tissue Imager in order to give reader a clear and overall idea about this instrument. In addition, the main limitation of this study, if any, should be described.

·      It is clear that this method was designed to perform immunofluorescence assays and to image fluorescent protein. However, most clinicians rely on histological sections to make diagnosis. Could The tissue imager be used also to image histological sections, such as H&E? Maybe it could be useful to show a comparison of the analysis of a histological sample acquired with tissue imager and conventional microscopy approaches.

·      I suggest to emphasize in the discussion section the importance of the new laboratory techniques compared to older approaches. At this regard I recommend to cite these articles, where the authors propose a new imaging approach compared to an older system. DOI: 10.1007/s00424-022-02686-8 ; 10.1155/2022/7908357 

Author Response

Please see attached response letter.

Round 2

Reviewer 1 Report

The revised manuscript is suitable for publication

Reviewer 3 Report

The authors have carefully addressed all my concerns, so I recommend the publication of this study.